# Regulatory B Lymphocytes Colonize the Respiratory Tract of Neonatal Mice and Modulate Immune Responses of Alveolar Macrophages to RSV Infection in IL-10-Dependant Manner

**DOI:** 10.3390/v12080822

**Published:** 2020-07-29

**Authors:** Daphné Laubreton, Carole Drajac, Jean-François Eléouët, Marie-Anne Rameix-Welti, Richard Lo-Man, Sabine Riffault, Delphyne Descamps

**Affiliations:** 1Université Paris-Saclay, INRAE, UVSQ, VIM, 78350 Jouy-en-Josas, France; daphne.laubreton@inserm.fr (D.L.); carole_drajac@orange.fr (C.D.); jean-francois.eleouet@inrae.fr (J.-F.E.); 2Université Paris-Saclay, UVSQ, Inserm, Infection et Inflammation, U1173, 78180 Montigny-Le-Bretonneux, France; marie-anne.rameix-welti@uvsq.fr; 3Laboratoire de Microbiologie, Hôpital Ambroise Paré, AP-HP, 92100 Boulogne-Billancourt, France; 4The Center for Microbes, Development and Health, Key Laboratory of Molecular Virology and Immunology, Unit Immunity and Pediatric Infectious Diseases, Institute Pasteur of Shanghai, Chinese Academy of Sciences, Shanghai 200031, China; richard.loman@ips.ac.cn; 5University of Chinese Academy of Sciences, Beijing 100039, China

**Keywords:** respiratory syncytial virus, neonates, age-dependent replication, lungs, innate immunity, immunoregulation, alveolar macrophage, regulatory B-cell, interleukine-10, interferons

## Abstract

Respiratory syncytial virus (RSV) is the prevalent pathogen of lower respiratory tract infections in children. The presence of neonatal regulatory B lymphocytes (nBreg) has been associated with a poor control of RSV infection in human newborns and with bronchiolitis severity. So far, little is known about how nBreg may contribute to neonatal immunopathology to RSV. We tracked nBreg in neonatal BALB/c mice and we investigated their impact on lung innate immunity, especially their crosstalk with alveolar macrophages (AMs) upon RSV infection. We showed that the colonization by nBreg during the first week of life is a hallmark of neonatal lung whereas this population is almost absent in adult lung. This particular period of age when nBreg are abundant corresponds to the same period when RSV replication in lungs fails to generate a type-I interferons (IFN-I) response and is not contained. When neonatal AMs are exposed to RSV in vitro, they produce IFN-I that in turn enhances IL-10 production by nBreg. IL-10 reciprocally can decrease IFN-I secretion by AMs. Thus, our work identified nBreg as an important component of neonatal lungs and pointed out new immunoregulatory interactions with AMs in the context of RSV infection.

## 1. Introduction:

Human respiratory syncytial virus (RSV) is the major cause of acute lower respiratory infections (LRI) in infants and young children, named bronchiolitis. RSV infection was estimated to be responsible for 48,000 to 74,500 hospital deaths among children under 5 years of age in 2015, with most deaths occurring in developing countries [1]. Moreover, there is a mounting body of evidence that severe LRI caused by RSV infection in early life are a risk factor for an increased incidence of asthma development [2,3]. The reasons for this neonatal hyper-susceptibility to severe RSV infection are still poorly described [4]. Understanding the specificities of neonatal immune responses to RSV infection is essential for the development of effective treatments and vaccines against the virus [5].

The immunological mechanisms associated with the early-life susceptibility to RSV infections and their long-term pathological consequences have begun to be deciphered using an experimental neonatal BALB/c mouse model. This experimental model showed the importance of age at first infection on the long-term immunopathological consequences [6]. Indeed, mice infected with RSV before seven days of life showed more severe weight loss, recruitment of inflammatory cells (eosinophils and neutrophils), and increased IL-4 production during reinfection in adulthood [7]. Moreover, RSV-infected neonatal mice have significant deficiencies in the pulmonary mobilization of antigen-presenting cells, dendritic cells (DCs), as well as in the production of type-I interferons (IFN-I) with antiviral properties [8,9,10,11]. Several publications have suggested an inadequate IFN-I production by DCs and demonstrated the impact of defective IFN-I response on the type-2-biased immunopathology to RSV infection and on the increased disease severity observed in neonates [8,9,10,11]. Conventional DCs (cDCs) and plasmacytoid DCs (pDCs) are not the only source of IFN-I since it has recently been shown in adult mice that alveolar macrophages (AMs) constitute the major source of IFN-I upon RSV infection [12]. However, the ability of neonatal AMs to produce IFN-I remains to be determined. Murine neonatal RSV infection is also characterized by a release of interleukine-33 (IL-33) into the lungs of mice, which is not the case in adult animals [13]. IL-33 is believed to play a major role in the immunopathogenesis of RSV infection in neonates by increasing the presence of type-2 innate lymphoid (ILC2) cells in the lungs, cells that have the ability to direct the immune response towards type-2 immunity [14]. These data support the hypothesis that infant susceptibility to RSV infection is intrinsically related to the immunological characteristics of the lung mucosa in the neonatal period, that promote a type-2 immunity in the developing lungs [4]. At homeostasis, compared to adult lungs, neonatal lung tissue (6 day-old BALB/c mouse) is poor in DCs and pDCs but enriched in CD4^−^ CD8^−^ lymphocytes expressing the transcription factor GATA3 which seem to be at the origin of type-2-biased neonatal immunity. Furthermore, conventional CD4^+^ and CD8^+^ T-cells, as well as B-cells are less represented than in adult [15].

A few years ago, a new population of innate cells was described in spleen of neonatal mice and defined as neonatal regulatory B CD5^+^ lymphocytes (nBreg) based on its IL-10-dependent immunoregulatory properties on type-1/type-2 polarization [16]. Indeed, following CpG stimulation, these nBreg produce large amount of IL-10 that limits TH1 priming capacity of neonatal conventional DCs by reducing their IL-12 production in response to a TLR9 ligand [16]. This early release of IL-10 by nBreg was shown to protect neonates from lethal inflammation and help to control the development of autoimmune diseases, such as experimental autoimmune encephalomyelitis [17,18]. nBreg originates from fetal liver and the neonatal liver input remains high during the first weeks of life [19]. A similar CD5^+^ nBreg population was characterized in human neonates and was shown to produce IL-10 in response to in vitro RSV infection [20]. Interestingly, clinical studies have shown that fatal outcomes of primary RSV infection disease is associated with massive pulmonary infiltrate of B-cells [21]. More particularly, the frequency of nBreg has been associated with a poor control of RSV infection and bronchiolitis severity in infants [20]. To date, although nBreg activity may constitute an early-life host response that favors RSV pathogenesis, their contribution on RSV disease remains poorly described.

In this paper, we show that the upper and lower respiratory tract of neonatal mice were particularly enriched in neonatal regulatory CD5^+^ B lymphocytes, and their presence correlated with the inability of neonates to produce IFN-I and inhibit viral replication. Furthermore, we demonstrate the capacity of neonatal AMs to mount potent IFN-I and inflammatory responses following in vitro RSV infection, albeit at a lower level than adult AMs. Finally, our work points out an original crosstalk between IL-10-producing nBreg and IFN-I-producing AMs in the context of neonatal RSV infection.

## 2. Materials and Methods

### 2.1. Reagents

Complete RPMI medium consisted of RPMI 1640 medium (Invitrogen) supplemented with l-glutamine 2 mM, 10% foetal calf serum (FCS, Eurobio, Les Ulis, France) and antibiotics (100 U/mL penicillin and 100 µg/mL streptomycin, ThermoFischer, Montigny Le Bretonneux, France). Uncomplete RPMI medium consist of RPMI 1640 medium (ThermoFischer) supplemented with l-glutamine 2 mM and antibiotics (100 U/mL penicillin and 100 µg/mL streptomycin, ThermoFischer). RSV-expressing luciferase (RSV-Luc; 1.4 × 10^7^ PFU/mL) or mCherry (RSV-mCherry; 1.4 × 10^7^ PFU/mL) were produced on HEp-2 cells (ATCC number CCL-23, Manassas, VA, USA) [22]. UV-inactivated-RSV consisted of RSV-mCherry exposed to 20 min ultra-violet (UV). Mock control consisted of HEp-2 cell culture supernatant.

### 2.2. Mice and Viral Infection

All experiments were approved by the local ethics committee COMETHEA (INRAE and AgroParisTech) under authorization number 15–16 (2015060414241349_v1, APAFIS#600, 24th July 2015) and were performed according to the European Community rules of animal care. Pregnant BALB/c mice were purchased from Janvier (Le Genest, St. Isle, France) were housed under Bio-Safety Level-2 conditions (IERP, INRAE, Jouy-en-Josas). Neonate (6 day-old or 14 day-old) mice received 10 µL of RSV-Luc or Mock by intranasal (in) instillation. Adult (6–8 week-old) mice received 50 µL of RSV-Luc Mock by in instillation [22]. Neonates were anesthetized by a brief cryoanesthesia [23], while adults were anesthetized by ketamine/xylazine (Alcyon, Landerneau, France) intraperitoneal (ip) injection (50 and 10 mg/kg respectively).

### 2.3. Sample Collection

Mice were euthanized by ip injection of pentobarbital (Alcyon, Landerneau, France, 50 µL for neonates and 150 µL for adults). Bronchoalveolar lavages (BAL) were performed by flushing the lungs via tracheal puncture two times with 250 µL (neonates) or 1 mL (adults) of Ca^2+^- and Mg^2+^-free PBS supplemented with 1 mM EDTA. BAL fluids were centrifuged 5 min at 250× *g* and supernatants were stored frozen at −20 °C. The lungs or nasal turbinates (NT) were collected in complete RPMI medium for flow cytometry analysis or directly frozen in nitrogen and conserved at –80 °C until processed for quantification of luciferase activity, RNA extraction or protein analysis. For flow cytometry or cell culture, the lungs, and NT were cut in small pieces and digested for 10 min with Collagenase D and DNase I (2 and 1 mg/mL, respectively). Tissue was then pressed on a 70 µm cell strainer using a syringe plunger to dissociate the cells. Cell strainer were washed with complete RPMI medium. Cells were centrifuged 10 min at 1200 rpm and suspended in PBS 2% FBS for flow cytometry staining. For total RNA extraction, the lungs were homogenized in 250 µL of lysis buffer (Nucleospin RNA kit, Macherey Nagel, Düren, Deutschland) with a Precellys 24 bead grinder homogenizer (Bertin Technologies, St Quentin en Yvelines, France) 1 × 15 s at 4 m/s. For luciferase activity or quantification of cytokines, the lungs or NT were homogenized in 300 µL of Passive Lysis Buffer (PLB; 1 mM Tris pH 7.9; 1 mM MgCl_2_; 1% Triton × 100; 2% glycerol; 1 mM DTT) with a Precellys 24 bead grinder homogenizer, 2 × 15 s at 4 m/s.

### 2.4. Viral N-RSV RNA Load and Gene Expression by Quantitative RT-PCR (qRT-PCR)

Total RNA was extracted from cell lysates using Nucleospin RNA kit (Macherey Nagel) and reverse transcribed using random primers and SuperScript II (ThermoFischer) according to the manufacturer’s instructions. The primers (Sigma–Aldrich, Saint Quentin Fallavier, France) used are listed in Table 1. qRT-PCR was run in triplicate for each gene using the MasterCycler R realplex (Eppendorf, Montesson, France) and SYBRGreen PCR Master Mix (Eurogentec, Seraing, Belgium). Fluorescence curves were analyzed using the Realplex software (Eppendorf) to determine the cycle threshold (Ct) values for each gene. Individual data were normalized to GAPDH mRNA, by calculating the ∆Ct (median Ct (gene) - median Ct (GAPDH)).

### 2.5. Bioluminescence Measurements

Photon emission was measured in the lungs and NT of RSV-Luc or Mock infected mice using the IVIS system (Xenogen Biosciences, Advanced Molecula Vision, Grantham, United Kingdom’s) and Living Image software (version 4.0, Caliper Life Sciences, Waltham, MA, USA). Briefly, mice received 50 µL of d-luciferin (30 mg/mL, Perking Elmer, Villebon-sur-Yvette, France) ip (neonates) or in (adults) and luciferase activity was measured for 1 min with f/stop = 1 and binning = 8. A digital false-color photon emission image of the mouse was generated, and photons were counted within a constant region of interest corresponding to the surface of the whole lungs or NT area. Results are expressed in radiance (photons/sec/cm^2^/sr). Photon emission was also analyzed in lung and NT homogenates by mixing 100 µL of lysate with 100 µL of d-luciferin (0,1 mg/mL) supplemented with ATP (0.5 M) in flat bottom 96-well black plates (Thermo Scientific, Bremen, Germany). Luciferase activity was measured for 1 min with f/stop = 1 and binning = 8. Radiance was normalized to the weight of tissues.

### 2.6. Cellular Culture and Infection

Primary alveolar macrophages (AMs) were isolated from BALs of neonates or adult mice, as previously described [24]. The purity and phenotype of AMs collected from neonatal and adult mice were described and were similar (Drajac et al., submitted). AMs (5 × 10^4^ cells/well) were plated in 96-well cell culture plates in complete RPMI medium for 24 h to allow adhesion. AMs were exposed at multiplicity of infection (MOI) = 5 to RSV-mCherry, UV-inactivated-RSV-mCherry, or Mock control for 2 h in uncomplete RPMI medium [22]. Then, medium was replaced with complete RPMI medium and cells were incubated for 24 h. In some experiments, complete RPMI medium was supplemented with recombinant murine IL-10 (0.5 mg/mL, PeproTech, Neuilly-sur-seine, France).

B-cells were positively enriched from pooled lung single cell suspensions using anti-CD19 biotinylated antibody (Biolegend, San Diego, CA, USA, #115504) and streptavidin-magnetic beads (Biolegend #480016) on magnetic MS MACS column (Miltenyi Biotec, Paris, France). B-cell subsets were then FACS-sorted on an ARIA cell sorter (Becton Dickinson, Franklin Lakes, NJ, USA) based on their CD19, CD23, and CD5 expression, to obtain CD5^+^, CD23^+^ and CD5^−^ CD23^−^ B-cell subsets. FACS-sorted B-cell subsets (5 × 10^5^ cells/well) were exposed at MOI 5 to RSV-mCherry, UV inactivated-RSV, or Mock control for 2 h in uncomplete RPMI medium, in the absence or presence of AMs (ratio AMs:B-cells = 1:5 or 1:10 corresponding to 5.10^3^ and 1.10^4^ AM/well respectively). Then, medium was replaced with complete RPMI medium and cells were incubated for 48 h. In some experiments (Figure 5D,E), B-cells were infected in the presence of supernatant from 24 h-infected AMs in the presence or absence of anti-IFNAR1 antibody (BioXcell, Lebanon, NH, USA, MAR1-5A3, 2.5 µµg/mL).

### 2.7. Flow Cytometry

After saturation with anti-CD32/CD16 (10 min, 4 °C), cells were incubated with surface antibodies listed in Table 2 for 20 min at 4 °C. Data were acquired with a FACSFortessa (BD biosciences, San Jose, CA, USA) and analyzed with the FlowJo Sofware v7.5 (Tree Star Inc., BD, Le Pont de Calix, France).

### 2.8. Measure of Cytokine Concentration by Multiplex Analysis

Cytokine and chemokine levels were measured from mouse BALs or lung homogenates using IFN-alpha/IFN-beta 2-Plex Mouse ProcartaPlex™ Panel and Cytokine & Chemokine 26-Plex Mouse ProcartaPlex™ Panel 1 (ThermoFisher Scientific), according to manufacturer instructions. Data were acquired using a Bio-Plex^®^ multiplex system (Bio-Rad). Results were analyzed using ProcartaPlexAnalyst software (ThermoFisher Scientific).

### 2.9. Statistical Analysis

Data were expressed as arithmetic mean ± standard error of the mean (SEM). Non-parametric Mann–Whitney (comparison of two groups, *n* ≥ 4) or *t*-test (comparison of two groups, *n* = 3) were used to compare unpaired values (GraphPad Prism software, San Diego, CA, USA). Significance is represented: * *p* < 0.05; ** *p* < 0.01; *** *p* < 0.001; and **** *p* < 0.0001.

## 3. Results

### 3.1. Regulatory CD5^+^ B Lymphocytes Colonize the Respiratory Tract of Neonatal Mice at a Very Early Stage

To analyze the activity of CD5^+^ B lymphocytes in the physiopathology of neonatal RSV infection, we first asked whether this population can be found in the respiratory tract of neonatal BALB/c mice. Thus, we characterized conventional (CD23^+^ CD5^−^), immature (CD23^−^ CD5^−^), and CD5^+^ (CD23^−^ CD5^+^) B lymphocyte subsets in upper (the nasal turbinates or NT) and lower (lungs) respiratory tract and in the spleen of BALB/c mice at various ages as compared to adult mice (gating strategy, Appendix A).

In adults, the lungs, the NT as well as spleen B-cell compartments contained almost exclusively CD23^+^ conventional B-cells, while their counterparts in neonates were enriched in CD23^−^ CD5^−^ immatures B-cells (Appendix A).

Over time, the frequency of CD23^−^ CD5^−^ immature B-cells in the respiratory tract decreased in favor of CD23^+^ conventional B-cells (Appendix A). Originally described in spleen of neonatal mice [16], CD5^+^ B-cells were identified in both the lungs and NT of BALB/c neonates (Figure 1A). Their abundance relative to age formed a bell-shaped curve with the highest percentage observed at age 6 days, when they accounted for almost 20% of the lung B-cell population, and 10% of NT or spleen total B-cells (Figure 1B and Appendix A). Then, the representation of the CD5^+^ B-cell subset quickly dropped to reach 3–5% of B-cells at age 21 days, a percentage still higher than in adults where CD5^+^ B-cells accounted for 1–2% of B-cells in all tissues. The number of CD5^+^ B-cells in the respiratory tract (the lungs and NT) followed the same patterns as described above, except that the peak of CD5^+^ B-cells in the respiratory tract was between 6 and 8 days of age (Figure 1C). The intensity of expression of CD5 on CD5^+^ B-cells also formed a bell-shaped curve with the highest intensity of expression observed at age 6 days (Figure 1D). Thus, similar to previous observation in spleen [16] (Appendix A), the neonatal respiratory B-cell compartment was enriched in CD5^+^ B-cell subset. In order to better characterize these cells, we looked for classical costimulatory molecule expression at steady state in 6 day-old BALB/c mice (Figure 1E). CD23^−^ CD5^−^ immature B-cells expressed low levels of co-stimulatory molecules or MHC II molecules, while PDL-1, CD40, CD80 and MHC II were highly expressed by CD23^+^ conventional B-cells. Interestingly, CD5^+^ B-cells were found to express higher levels of all of these molecules than the CD23^+^ B-cells.

The CD5^+^ B-cells previously identified in the spleen of neonatal mice are known to have an immunosuppressive activity through the production of interleukine-10 (IL-10) [16]. Thus, different neonatal B-cell subsets were FACS sorted and exposed for 48 h to RSV. First, RSV replication measured by the level of the viral *N-RSV RNA* expression was detected in all B-cell subsets (Figure 1F). These data confirmed that neonatal CD5^+^ B-cells are permissive to RSV infection, as previously described in humans [20]. Interestingly, only the neonatal CD5^+^ B-cells were able to express *IL-10 RNA* (Figure 1G) and to produce high amount of IL-10 in response to RSV infection (Figure 1H). The induction of IL-10 was abrogated by UV-inactivation of RSV (UV-inactivated RSV or Inac. RSV) suggesting that live virus was required for IL-10 production by neonatal CD5^+^ B-cells. No other cytokines were detected in supernatants of RSV-infected neonatal CD5^+^ B-cells (no detection with cytokine and Chemokine 26-plex Mouse ProcartaPlex Panel1 assay, Thermofisher). Altogether, these data demonstrate that the CD5^+^ B-cell population is highly specific to the neonatal period and has the capacity to make IL-10 upon exposure to live RSV. Based on their characteristics similar to the regulatory B-cells previously described in human neonates [20], we named these cells neonatal regulatory CD5^+^ B lymphocytes (nBreg).

### 3.2. Neonatal Mice Are Highly Permissive to RSV Infection in Combination with Inadequate Type-I Interferon (IFN-I) Production

More recently, a high frequency of nBreg has been correlated with a poor control of RSV infection in human newborns and with bronchiolitis severity [20]. We took advantage of the availability of luciferase reporter RSV (RSV-Luc [22]) to perform a comprehensive in vivo bioluminescent imaging follow-up of the viral replication and spread according to the age of infection in BALB/c mice. Thus, 6 day-old or 14 day-old neonates and 8 week-old adult mice were infected with RSV-Luc. Then, in vivo bioluminescence intensity corresponding to RSV replication was determined daily (Figure 2A). As previously described [22], bioluminescence was observed in NT and lungs of adult mice as early as 2 days post-infection (d.p.i.). Maximum viral replication was detected at 4 d.p.i. in the lungs and remained detectable at 5 d.p.i., as shown by quantification on luciferase activity in living mice (Figure 2B) or in lung and NT homogenates (Figure 2C). The restrained viral production during the first days of infection in adult mice is dependent upon IFN-I-dependent antiviral response [9,12]. Thus, the production of IFN-α and IFN- β in adult mice was detected by luminex assays at 1 d.p.i. in lung homogenates (Figure 3) and from 1 d.p.i. to 3 d.p.i. in bronchoalveolar lavages (BAL; Appendix A). In contrast, RSV replication in 6 day-old neonates was immediately visualized in the respiratory tract (Figure 2A). Maximum bioluminescence was observed on day 1 and 2 in the lungs and decreased at 3 d.p.i. but remained detectable at 4 d.p.i. (Figure 2B). In NT of 6 day-old mice, RSV replication reached its maximum on 2 d.p.i. and remained strong until 5 d.p.i. (Figure 2A–C). In agreement with previous studies showing limited and inadequate IFN-I responses in RSV-infected neonates [8,9], IFN- α and IFN- β proteins were not detectable at 1 d.p.i. in lung homogenates (Figure 3) and from 1 d.p.i. to 3 d.p.i. in BAL (Appendix A). Moreover, CD5^+^ B-cell percentages in the respiratory tract were unaffected by RSV infection in 6 day-old BALB/c mice (Appendix A). Fourteen day-old neonates infected with RSV-Luc displayed a viral replication kinetic similar to RSV-infected adult mice (Figure 2A–C), together with a moderate IFN-I response, as illustrated by the detection of IFN- β proteins at 1 d.p.i. in lung homogenates (Figure 3). Thus, between six and 14 days of life, mice became capable of containing the viral replication at early d.p.i. and at the same period of life starts to produce small amount of IFN- β, confirming an age-dependent anti-viral response to RSV infection [8,9]. Interestingly, this age period was associated with a decrease in CD5^+^ nBreg numbers in the lung of neonate, to reach numbers close to those of adults (Figure 1A). Taken together, these data suggest that the neonatal period of life is characterized by a permissiveness to RSV replication, that is progressively lost with age in relationship with an increasing capacity to mount IFN-I responses. This phenomenon may also be related to the decrease in CD5^+^ nBreg rates in the airways. We therefore sought to investigate how these cells might interact with the antiviral response in this neonatal period. We focused our study on 6 day-old mice, a period when the CD5^+^ nBreg are still very abundant in the lungs of the animals.

### 3.3. Neonatal Primary Alveolar Macrophages (AMs) Are Able to Mount Anti-Viral and Inflammatory Responses upon In Vitro RSV Infection

It has been demonstrated in transgenic adult *Ifna6^gfp/+^* mice that AMs are the major source of IFN-I in response to RSV infection [12]. Since the neonatal IFN-I response to RSV infection is barely detectable in vivo [8,9,10], we postulated that AMs in a lung-neonatal context are not able to mount a potent IFN-I response to RSV infection. To address this issue, primary AMs isolated from BAL of 6 day-old or adult BALB/c mice were infected in vitro with RSV and the IFN-I and inflammatory responses were evaluated both by qRT-PCR on cell lysates and luminex assays in supernatants. Neonatal AMs expressed *IFN-α1* and *IFN-β mRNA* (Figure 4A) and produced detectable quantities of IFN-α and IFN-β in response to RSV infection (Figure 4B), however these IFN-I responses were significantly lower than in adult AMs (Figure 4B). Conversely, higher level of *N-RSV RNA* was detected in neonatal than in adult AMs (Figure 4C). Lower inflammatory cytokine responses (IL-6 and TNF-α) were measured in neonatal compared to adult AMs (Figure 4D). RSV can induce IL-10 production by human adult AMs [25]. In our experimental set-up for detection of inflammatory cytokines (Figure 4), we never detected IL-10 production by RSV-infected neonatal and adult AMs (no detection with luminex assays on cellular supernatants and qPCR on cell lysates). Interestingly, these anti-viral and inflammatory responses were abrogated upon exposure to UV inactivated-RSV suggesting their dependence on viral replication in AMs. In conclusion, albeit at lower levels than the adult responses, primary neonatal AMs from BALB/c mice are able to mount in vitro significant anti-viral and inflammatory responses to RSV infection. Thus, the pulmonary microenvironment of neonates may generate a tissue-specific context that renders neonatal AMs incapable of making an antiviral response upon RSV infection.

### 3.4. IFN-I Produced by Neonatal Primary AMs Enhance the Secretion of IL-10 by CD5^+^ nBreg upon Infection with RSV

We next evaluated to what extent different B-cell subsets co-cultured in vitro with neonatal AMs could impact their anti-viral and inflammatory responses to RSV infection. Neonatal AMs (1 × 10^4^ cells) were co-cultured and infected with RSV in presence of CD5^+^ nBreg, CD23^−^ CD5^−^ immature- or CD23^+^ conventional B-cells (ratio AMs:B-cells = 1:5). The production of both IFN-α and IFN-β as well as of IL-6 and TNF-α was significantly increased when AMs were co-cultured with CD23^+^ conventional B-cells in comparison with the condition AMs alone (Figure 5A,B). Co-culture of AMs with CD23^−^ CD5^−^ immature-B-cells resulted in a moderate increase of all cytokines, significant for IFN-α, IL-6, and TNF-α (Figure 5A,B). Finally, the co-culture of neonatal AMs with CD5^+^ nBreg led to a moderate increase in IFN-I, significant only for IFN-α (Figure 5A) but not for IFN-ββ, the IFN-I molecule most strongly produced by adult and neonatal AMs following exposure to RSV (see Figure 4B). IL-6 and TNF-α productions were not modified when AMs were exposed to RSV in presence of CD5^+^ nBreg (Figure 5B). Thus, CD23^−^ CD5^−^ immature and CD23^+^ conventional B-cell subsets enhanced the anti-viral and inflammatory responses of neonatal AMs upon RSV infection in vitro. Such increased activation of neonatal AMs was not triggered by CD5^+^ nBreg, the population of B-cell subsets which was found particularly abundant in the respiratory tract of 6 day-old neonates and remained stable throughout an RSV infection.

In a mouse model of acute inflammation, IFN-α and IFN-β were found to be critical for IL-10 production by neonatal CD5^+^ B-cells [26]. First, we investigated whether the response of CD5^+^ nBreg to an in vitro RSV infection could be modulated by IFN-I produced by neonatal AMs. To address this question, different neonatal B-cells subsets were co-cultured with neonatal AMs (1 × 10^4^ cells) and infected with RSV or UV inactivated-RSV (with ratio AMs:B-cells = 1:5 or 1:10, the latter ratio roughly matching the proportions of these cells in the newborn’s lungs). The production of IL-10 was measured in the supernatant of the co-cultures by luminex assay. AMs alone did not produce any detectable IL-10 when exposed to RSV, whether infectious or UV-inactivated (Figure 5C). The basal level of IL-10 production by CD5^+^ nBreg exposed to infectious RSV was increased 3–5-folds when neonatal AMs were added, in a dose-ratio dependent manner (Figure 5C). CD23^−^ CD5^−^ immature and CD23^+^ conventional B-cells remained unable to produce IL-10 upon RSV infection even in the presence of neonatal AMs whatever the ratio between cells. Second, we investigated whether IFN-I produced by RSV-infected AMs was driving the amplified IL-10-production of CD5^+^ nBreg. To address this question, the supernatant from RSV-infected neonatal AMs was collected. Although AMs are permissive to RSV infection, it does not allow the production of viral particles [27]. Thus, this conditioned medium was expected to be virus-free. CD5^+^ nBreg were then exposed to RSV in the presence of this conditioned medium, with or without IFNAR blocking antibody. IL-10 production by CD5^+^ nBreg tended to increase in the presence of conditioned medium (Figure 5D) while RSV replication, measured by the level of the viral *N-RSV RNA* expression tended to diminish, although statistical analysis could not be performed due to small number of experimental data points (Figure 5E). These effects were abrogated in the presence of IFNAR blocking antibody (Figure 5D,E). Thus, upon RSV infection in vitro, IFN-I produced by neonatal AMs promoted IL-10 production from CD5^+^ nBreg.

### 3.5. IL-10 Exerts Strong Modulatory Effects on the Response of Neonatal AMs to RSV Infection

When exposed to RSV in vitro, CD5^+^ nBreg produced IL-10, a cytokine known to exert an immunosuppressive activity on its immediate environment [28]. Thus, we postulated that IL-10 may modulate the capacity of neonatal primary AMs to react to an RSV infection. Neonatal AMs were infected with RSV in presence of recombinant mouse IL-10, and then IFN-I, chemokine, and cytokine responses were analyzed by luminex assay in supernatants (Figure 6A–C). When neonatal AMs were infected with RSV in presence of IL-10, their capacity to produce IFN-α and IFN-β was significantly reduced but not totally abrogated compared to the condition RSV alone (Figure 6A). Similarly, a significant reduction of CCL3 and CCL5 chemokines (Figure 6B) as well as TNF-α was observed when IL-10 was added during the course of RSV infection. The IL-10 treatment did not change the capacity of RSV-infected AMs to produce IL-6 but it increased very significantly their capacity to make CCL7 and CCL11 chemokines and different cytokines, such as IL-17A, IL-22, IL-23, and IL-27 (Figure 6B,C). Thus, IL-10 induces the conversion of neonatal AMs from an antiviral phenotype to a new pro-type 17 immune response phenotype.

## 4. Discussion

CD5^+^ nBregs have been associated with bronchiolitis severity in human newborns [20]. However, their presence in the lungs of neonates and their contribution to RSV disease remain poorly described. In the present study, our results showed that CD5^+^ nBreg are particularly abundant in the respiratory tract of murine neonates at an age period corresponding to a window of higher permissiveness to RSV replication and a restrained IFN-I response. While no IFN-I was detected in the lungs of RSV-infected neonates, we showed that neonatal primary AMs were able to mount potent IFN-I and inflammatory responses upon in vitro RSV exposure, and this IFN-I production strengthened IL-10 secretion by CD5^+^ nBreg.

In order to understand the susceptibility of infants to severe RSV infection, a mouse model of neonatal infection has been developed [7]. Neonates infected at the age of 7 days develop an asthma-like pathology upon adult reinfection that recapitulates the symptoms observed in human infants. We and others showed that while *RSV-N RNA* [9] or *RSV-L RNA* [7] loads were lower in infected neonates than in adults at all-time points, they followed a similar kinetic. Using luciferase-expressing RSV, we were able to conduct a comprehensive follow-up of the viral replication in vivo using bioluminescent imaging. We showed that RSV infection behaved differently in a neonatal environment as compared to adult. Indeed, RSV-Luc replication in 6 day-old infected neonates started as early as 1 d.p.i. and was still detectable at 4 d.p.i., especially in the NT, while in adults, RSV replication started to be detectable at a low level at 2 d.p.i. to reach its maximum at 4 d.p.i.. This higher neonatal sensitivity to early RSV replication correlated with their inability to mount potent IFN-I response. The respiratory immune compartments of neonates and adults differ in many aspects. After birth, AMs accumulate in the lungs until 3 days of age, which coincides with the start of the alveolarization phase [29]. This phase is characterized by a recruitment wave of type-2 innate cells (ILC2s, mast cells, eosinophils, and basophils) that reach their maximum at 14 days of age and then decline until weaning (day 21) [4,30,31]. Finally, DCs, T- and total B-cells accumulate over time in the lungs until weaning [4,30]. We observed similar immune cell colonization kinetics in the NT of BALB/c mice (unpublished data). Interestingly, despite the fact that their respiratory immune compartment is still characterized by an enrichment in type-2 cells and low numbers of T- and B-cells, 14 day-old neonates showed similar RSV replication kinetic to adults and were able to produce IFN-I, even if the levels remained lower than in adults. Thus, between 6 and 14 days of life, mice become capable to control the viral replication at early d.p.i. and developed an anti-viral response confirming an age-dependent production of IFN-I upon RSV infection [8,9,10].

In this paper we identified a CD5^+^ B-cell population that accumulated in the lungs and the NT of neonates to reach its maximum between 6 and 8 days of age, when these CD5^+^ B-cells displayed a pre-active state characterized by high expression of co-stimulatory molecules. This population then disappear progressively with age. Similar CD5^+^ B-cell population, named nBreg, was previously observed in human newborns and was shown to produce IL-10 in vitro in response to RSV exposure [20]. Our data demonstrated that neonatal murine respiratory CD5^+^ B-cells also produced IL-10 following in vitro RSV infection. We thus named these cells CD5^+^ nBreg. Interestingly, we observed that the accumulation of CD5^+^ nBreg in the neonatal lungs was associated with an increase in the CD5 expression at their cell surface until the age of 5 days (see Figure 1D). The role of CD5 on B-cells is not completely establish but it could play a role on B-cell survival through regulation of IL-10 production [32,33].

The period of life when IL-10-producing CD5^+^ nBreg are most present in the lungs parallels the early life period with poor anti-viral responses [8,9]. Different immune regulatory functions of IL-10 have been described [28], but the role of IL-10 in the pathophysiological mechanisms of RSV infection in the newborn is not yet clearly established. High concentrations of IL-10 are found in nasopharyngeal secretions and in sera of children with RSV infection [34,35]. Therefore, we postulated that CD5^+^ nBreg-derived IL-10 could be responsible for the limited IFN-I in neonates. Different innate cell populations, such as pDCs, DCs, or AMs, are able to mount an anti-viral response following RSV infection and produce IFN-I [10,11]. Both pDCs and DCs are poorly represented in neonatal lungs [8,9] whereas AMs, which have been identified as the main source of IFN-I upon RSV infection in adult mice, have already fully colonized the alveolar space in 6 day-old neonates [29]. We demonstrate for the first time that neonatal AMs were able to mount in vitro a potent IFN-I response as well as inflammatory response to RSV infection, although lower than the one of adult AMs. In contrast to human adult AMs [25], RSV-infected AMs from adult or neonatal mice did not make detectable amount of IL-10. We show that the production of IFN-β, IL-6 or TNF-α by neonatal AMs was boosted by B-cell subsets, CD23^−^CD5^−^ immature and CD23^+^ conventional B-cells. Interestingly, this was not observed with IL-10-producing CD5^+^ B-cells that colonize the neonatal lungs. However, co-culture experiments with amounts of AMs and/or nBreg used per well were not sufficient to reveal a strong activity of IL-10 on AM anti-viral response. Nevertheless, exogenous IL-10 repressed both the IFN-I and inflammatory responses (TNF-α and inflammatory chemokines CCL3 and CCL5) of neonatal AMs, while it increased the secretion monocyte and eosinophil attracting chemokines, such as CCL7 and CCL11 [36]. Macrophages are known to have phenotypic and functional plasticity in order to appropriately respond to a variety of stimuli in their environment [37,38]. The AM phenotype is thus particularly sensitive to its microenvironment [39]. We observed that exposure of neonatal AMs to IL-10 induced such a plasticity in secreted cytokines, like IL-17A, IL-22, IL-23, and IL-27 in response to RSV infection. This IL-10-driven polarization may be important to reduce tissue damage [37,38]. In particular, IL-27 is reported to control the balance of protective and pathogenic T-helper (TH) cell subsets during RSV infection [40,41]. In a model of CpG-induced-inflammation, DCs-derived IFN-I help to control inflammation by increasing the capacity of splenic CD5^+^ nBreg to produce IL-10 which negatively regulate both IFN-I and inflammatory responses [26]. In order to evaluate if such regulatory loop between respiratory CD5^+^ nBreg and AMs could be involved in response to RSV, we co-infected AMs and the various subsets of B-cells for 48 h with RSV. As expected, IL-10 production by CD5^+^ nBreg was increased in the presence of AMs in an IFN-I dependent manner, since this effect was abrogated in the presence of anti-IFNAR antibody. Using a model of lung-specific IL-10 overexpression (OE) mice, Sun et al., showed that depending on the timing of IL-10 overexpression during the course of RSV infection, IL-10 could either dampen inflammation or exacerbate RSV immunopathology thus demonstrating a dual role for IL-10 on the immune response to RSV [42]. Thus, CD5^+^ nBreg-derived IL-10 could not only modulate inflammatory responses, but also influence T-helper cell polarization. In mice, it was shown that IL-10 production by splenic CD5^+^ nBreg following TLR9 stimulation prevent TH1 priming by limiting the IL-12 production of neonatal DCs [42]. In human newborns, CD5^+^ nBreg-derived IL-10 was shown to dampened TH1 cytokine production [20]. Finally, adult IL-10-deficient mice showed greater IFN-γ response to RSV [43]. The impact of IL-10 production by respiratory CD5^+^ nBreg on the type of T-cell response elicited by RSV in murine neonate remains to be evaluated. In vivo, RSV infection did not impact the number of CD5^+^ nBreg in the lungs or the NT. Although IL-10 was secreted in the supernatant of CD5^+^ nBreg exposed to RSV, IL-10 was not detectable in the respiratory tract or BAL of 6 day-old infected neonates or adults (no detection with Luminex assays). Still, lung explants from 6 day-old neonates made detectable amounts of IL-10, at greater levels than adult lung explants (data not shown, Laubreton et al. [44]), supporting the hypothesis of an in vivo IL-10 response to RSV by the neonatal lung tissue.

Finally, IFN-I is not the only modulator of CD5^+^ nBreg function. IL-33, an alarmin that belongs to the IL-1 family, is rapidly secreted following RSV exposure in neonates and contributes to the promotion of type-2 responses by promoting IL-13 and IL-5 production by ILC2 [13,45]. Recently, the importance of ILC2 in shaping the immune response early during RSV infection has been revealed in infants [46]. AMs are one of the source of IL-33 upon RSV infection [45,47,48], and it was shown that deletion of IL-33 production by myeloid cells strongly affects IL-5 secretion and eosinophil recruitment upon RSV infection [45]. Furthermore, IL-33 treatment promotes IL-10 production by B-cells in adult mice, thus protecting these mice from induction of inflammatory bowel disease [49]. RSV-induced IL-33 in neonate lungs could participate in the promotion of IL-10 production by CD5^+^ nBreg. Future studies will evaluate the role of the IFN-I/IL-33 balance on the transition from deleterious neonatal to safe adult responses to RSV infection.

## 5. Conclusions

In conclusion, our work identified CD5^+^ nBreg as an important component in early innate immunity of neonatal lungs. Moreover, our data highlighted a crosstalk between CD5^+^ nBreg and neonatal AMs in the context of neonatal RSV infection and pointed out once again that the neonatal susceptibility to RSV infection is intrinsically linked to the immunological characteristics of young pulmonary mucosa.

## Figures and Tables

**Figure 1 viruses-12-00822-f001:**
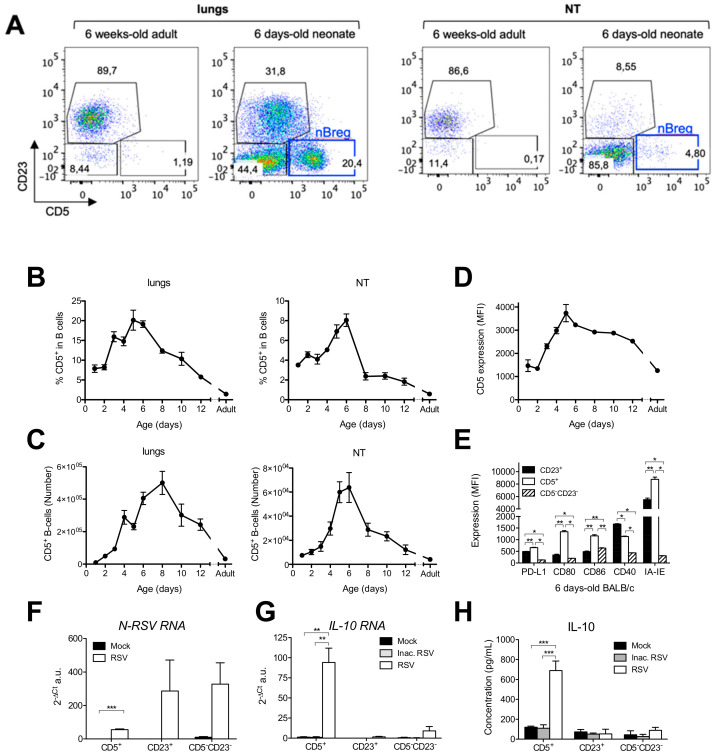
Neonatal regulatory B lymphocytes accumulated in the respiratory tract during the early period of life and produce IL-10 in response to RSV infection. The lungs and nasal turbinates (NT) were collected from BALB/c mice at various age (*n* = 5–6 neonates/age, and 9 adults) and analyzed for B-cell populations by flow cytometry. (**A**) Conventional (CD23^+^ CD5^−^), immature (CD23^−^ CD5^−^), and CD5^+^ B-cells were separated according to their CD23 and CD5 expression. Representative pseudocolor dot plot of CD23/CD5 staining on B-cells of adult and 6 days-old are shown. (**B**–**D**) Age evolution of CD5^+^ B-cell subset percentage among CD19^+^ (**B**) number (**C**), as well as CD5 expression intensity on pulmonary CD5^+^ B-cells (**D**) are reported. (**E**) Expression levels of various activation markers were compared between B-cell subsets. Results are expressed as mean of individual ± SEM (one representative experiment of two). The statistical significance of differences was determined by the Mann–Whitney test (** *p* < 0.01). (**F**–**H**) CD5^+^, CD23^+^, and CD5^−^ CD23^−^ from the lungs of 6 day-old BALB/c mice (pool of *n* = 50) were FACS sorted and infected for 48 h with RSV or UV-inactivated RSV (Inac. RSV, 5 × 10^4^ cells/well, MOI = 5) or Mock control. (**F**) Infection was analyzed by measurement of viral *N-RSV* RNA expression by q-RT-PCR from cell lysates. (**G**,**H**) IL-10 production was measured q-RT-PCR on cell lysates (**G**) and by luminex assays on cellular supernatants (**H**). Results are expressed as mean of triplicate ± SEM (one representative experiment of three). The statistical significance of differences was determined by unpaired *t*-test (* *p* < 0.05; ** *p* < 0.01; *** *p* < 0.001).

**Figure 2 viruses-12-00822-f002:**
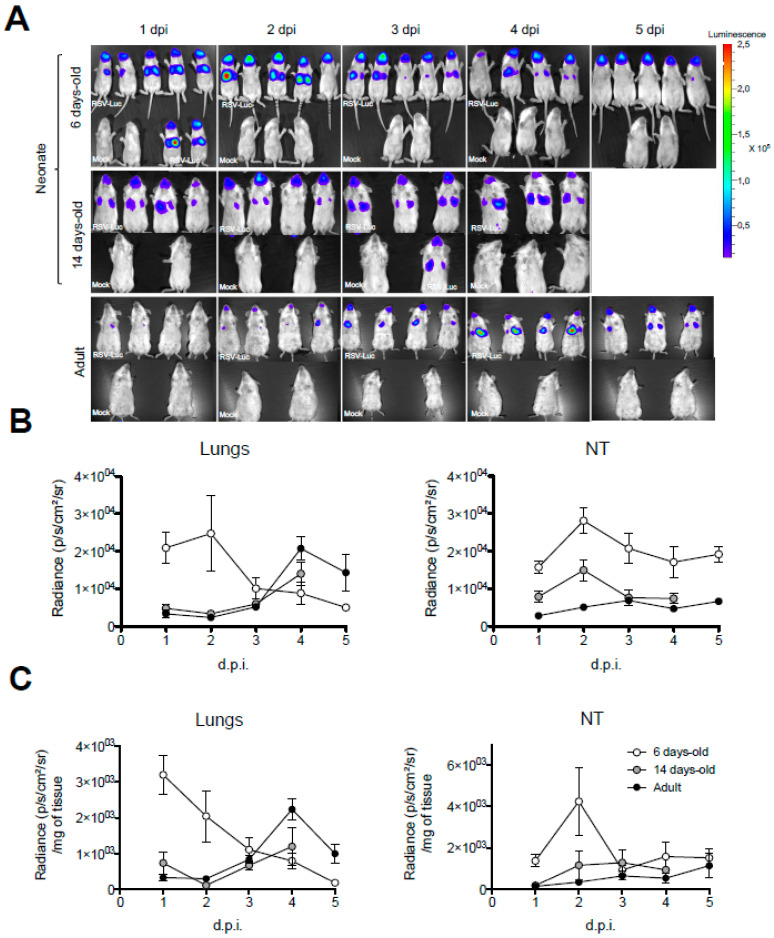
Age-dependent RSV replication kinetic in BALB/c mice. Six day-old or 14 day-old neonates and 8 week-old adult mice were infected i.n. with RSV-Luc (1.75 × 10^6^ pfu/mL, 10 µL/neonate, 50 µL/adult). (**A**) At each time point after infection, mice received luciferin (1.5 mg under 50 µL, administrated ip for neonates and in for adults) and luciferase activity was visualized in the IVIS-200 imaging system. (**B**) Quantification of luciferase activity in the lungs and NT of living mice (radiance in photon/s/cm^2^/sr) or (**C**) in lung and NT homogenates (radiance in photon/s/cm^2^/sr/mg of tissues). Results are expressed as mean of individual ± SEM (one representative experiment of two with *n* = 5 to 8 mice/group).

**Figure 3 viruses-12-00822-f003:**
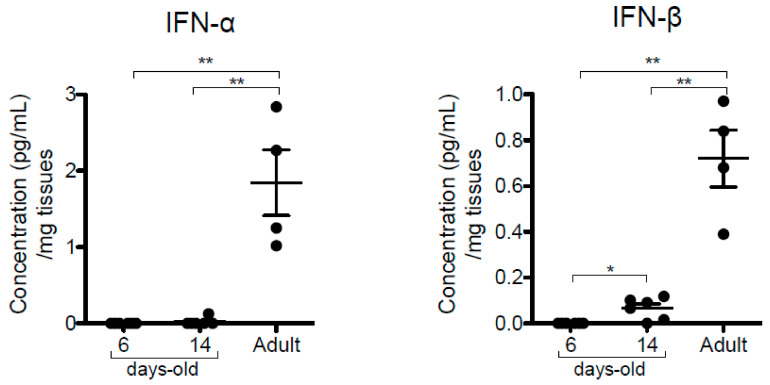
Type-I interferon (IFN-I) antiviral response was not detectable in neonates. Six day-old or 14 day-old neonates and 8 week-old adult mice were infected with RSV-Luc (1.75 × 10^6^ pfu/mL, 10 µL/neonate, 50 µL/adult). IFN-α and IFN-β production was measured by luminex assays in lung homogenates at 1 d.p.i. Results are expressed as mean of individual ± SEM (one representative experiment of two with *n* = 5 to 8 mice/group; * *p* < 0.05; ** *p* < 0.01).

**Figure 4 viruses-12-00822-f004:**
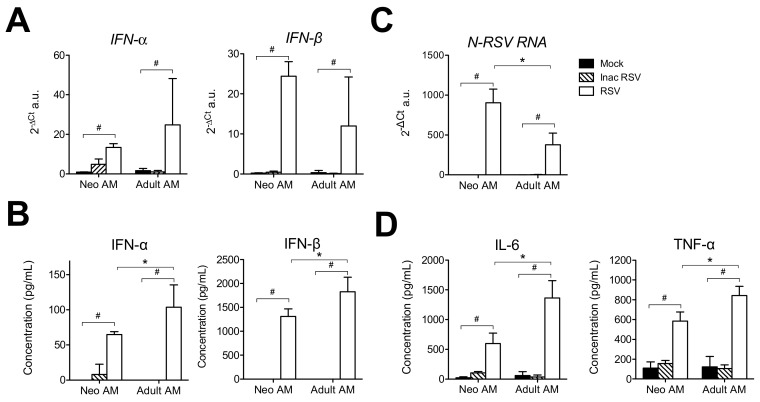
Neonatal primary AMs produced IFN-I and inflammatory cytokines in response to in vitro RSV infection. Primary AMs of 6 day-old (pool of *n* = 50) or adult (pool of *n* = 5) BALB/c mice were exposed for 24 h to RSV-mCherry, UV inactivated-RSV-mCherry (Inac RSV, 5 × 10^4^ cells/well, MOI = 5), or Mock control. Type-I IFN responses were evaluated (**A**) by qRT-PCR on cell lysates and (**B**) by luminex assays in cellular supernatants. (**C**) Infection was analyzed by measurement of viral *N-RSV RNA* expression by qRT-PCR from cell lysates. (**D**) Inflammatory cytokine production in supernatants was evaluated by luminex assays. Results are expressed as mean of triplicate ± SEM (one representative experiment of three). # indicated the statistical significance of differences between control conditions and RSV condition and was determined by the Mann–Whitney test (# *p* < 0.05). The statistical significance of differences between neonatal and adult primary AMs was indicated by * symbol and was determined by the Mann–Whitney test (* *p* < 0.05).

**Figure 5 viruses-12-00822-f005:**
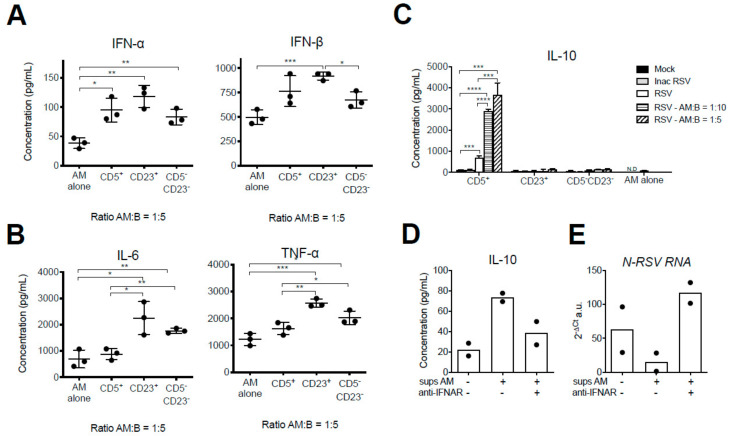
The IFN-I response of neonatal AMs to RSV enhanced IL-10 production by CD5^+^ B-cells. (**A**,**B**) Neonatal primary AMs (1 × 10^4^ cells/well, MOI = 5) were infected for 48 h with RSV-mCherry alone or in the presence of FACS-sorted CD5^+^, CD5^−^ CD23^+^, or CD5^−^ CD23^−^ B-cells (5 × 10^4^ B-cells/well, ratio AMs:B-cells = 1:5) isolated from the lungs of 6 day-old BALB/c mice (pool of *n* = 50). (**A**) IFN-α and IFN-β, and (**B**) IL-6 and TNF-α productions were determined by luminex assays in cellular supernatants. (**C**) CD5^+^, CD5^−^ CD23^+^, or CD5^−^ CD23^−^ B-cells isolated from the lungs of 6 day-old BALB/c mice (5 × 10^4^ cells/well, pool of *n* = 50) were infected for 48 h with RSV-mCherry, UV inactivated-RSV-mCherry (Inac RSV, MOI = 5), or Mock control, alone or in the presence of neonatal primary AMs at different ratios (5 × 10^3^ or 1 × 10^4^ AMs/well, ratio AMs:B-cells = 1:10 or 1:5). IL-10 production was measured by luminex assays in cellular supernatants. (**D**–**E**) FACS sorted CD5^+^ B-cells were infected for 48 h in the presence of conditioned supernatant from RSV-infected neonatal primary AMs in the presence or absence of anti-IFNAR antibody. (**D**) IL-10 secretion was measured in supernatant by luminex. (**E**) Infection was analyzed by measurement of viral *N-RSV RNA* expression by qRT-PCR from cell lysates. Results are expressed (**A**,**D**,**E**) mean of triplicate ± SEM (one representative experiment of three) or (**B**,**C**) as mean of duplicate (one experiment). The statistical significance of differences was determined by unpaired *t*-test (* *p* < 0.05; ** *p* < 0.01; *** *p* < 0.001).

**Figure 6 viruses-12-00822-f006:**
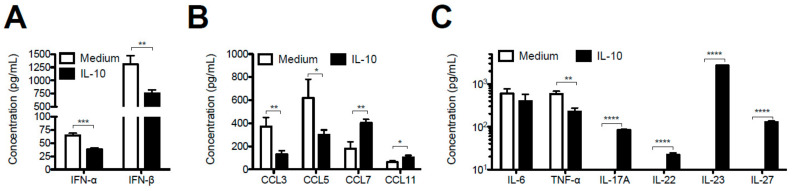
IL-10 altered the response of neonatal AMs to in vitro RSV infection. Neonatal AMs of 6 day-old neonates (pool of *n* = 50) were infected for 24 h with RSV (5 × 10^4^ cells/well, MOI = 5) in the presence of recombinant mouse IL-10 (0.5 mg/mL). (**A**) IFN-α and IFN-β, (**B**) chemokine or (**C**) cytokine productions were measured by luminex assays in cellular supernatants. Results are expressed as mean of triplicate ± SEM (one representative experiment of two). The statistical significance of differences was determined by unpaired *t*-test (* *p* < 0.05; ** *p* < 0.01; *** *p* < 0.001; **** *p* < 0.0001).

**Table 1 viruses-12-00822-t001:** Primer sequences.

Name of Gene	Forward Primer (5′ to 3′)	Reverse Primer (5′ to 3′)
*GAPDH*	GGGGTCGTTGATGGCAACA	AGGTCGGTGTGAACGGATTTG
*N (huRSV-A2)*	AGATCAACTTCTGTCATCCAGCAA	TTCTGCACATCATAATTAGGAGTATCAAT
*IL-10*	GCTGGACAACATACTGCTAACC	ATTTCCGATAAGGCTTGGCAA
*IFNα1*	GAGAAGAAACACAGCCCCTG	TCAGTCTTCCCAGCACATTG
*IFNβ*	CCCTATGGAGATGACGGAGA	CTGTCTGCTGGTGGAGTTCA

**Table 2 viruses-12-00822-t002:** List of antibodies.

Fluorochrome	Antibody	Clone	Supplier	Reference	Lot
APC	B220	RA36B2	Biolegend	103212	B189921
PerCPCy5.5	CD3	17A2	Biolegend	100218	B188934
PECy7	CD4	RM4-5	Biolegend	100528	B173316
PE	CD5	53-7.3	BD	553022	3343749
BV421	CD5	53-7.3	SONY Biotechnology, San Jose, CA, USA	100617	1103090
APC	CD19	1D3	BD	550992	38354
FITC	CD23	B3B4	BD	553138	3234818
PECy7	CD23	B3B4	SONY Biotechnology	101614	99531
PE	CD40	3/23	BD	553791	31674
APC-Cy7	CD45.2	104	Biolegend	109824	B205513
FITC	CD80	16-10A1	BD	553768	48602
PE	CD86	GL1	BD	553692	05843
FITC	IA-IE	2G9	BD	553623	4237594

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
