# Peer review of "Regulatory B Lymphocytes Colonize the Respiratory Tract of Neonatal Mice and Modulate Immune Responses of Alveolar Macrophages to RSV Infection in IL-10-Dependant Manner"

_viruses, 2020, doi:10.3390/v12080822_

Round 1
Reviewer 1 Report
The authors present clear evidence of the population of CD5+ B Ly cells in the upper and lower respiratory tract, in vivo and in vitro arguments for the importance of this cell population in the pathogenicity of RSV in a neonatal mouse model, present arguments for the permisivity of RSV infection due to these cells in neonate mice and not in adult mice.
Minor revisions and suggestions I would like to add/highlight are:
- In Figure 2, panel A, I find it uncertain why panels present different numbers of animals (2 vs 3 mock ,7 vs 4 RSV-Lu, etC). I would also like to ask why there is no 5 d.p.i. data for the 14 days-old mice group.
- While I appreciate you describe the level of significance in the text, it is good practice to also include this information in the figure description, so that it is handy for the reader.
- I think the use of "MOI" should be extensively presented in the text
- Line 161: The authors are asked to clarify "In some experiment"
- Line 429: showed similar RSV replication kinetic than adults and were able to produce - perhaps change to "to"; ".. similar .. to ..".
Author Response
- In Figure 2, panel A, I find it uncertain why panels present different numbers of animals (2 vs 3 mock ,7 vs 4 RSV-Lu, etC).
The number of individuals that can be placed on the field area to be imaged is related to the size of the animal at a given age. This is why we can display higher number of 6-days-old neonates than adults in a picture. We always put more RSV-infected than mock-infected individuals on one image (to illustrate the variability with infection that does not occur with controls, no signal). These panels are shown to illustrate the graphs with all individuals per group (panels B, C).
- I would also like to ask why there is no 5 d.p.i. data for the 14 days-old mice group.
The experiment with 14-days-old mice was conceptualized to illustrate the loss of early permissiveness of neonates to RSV infection. In this perspective of studying early infection and in an effort to limit the use of animals in the experiments, we did not plan analyses at 5 days post-infection, a time when lung replication disappears in both adults and 6-days-old mice.
- While I appreciate you describe the level of significance in the text, it is good practice to also include this information in the figure description, so that it is handy for the reader.
Figures are corrected as suggested.
- I think the use of "MOI" should be extensively presented in the text
Line 149, the mention MOI was corrected as “ Multiplicity of Infection (MOI) = 5”
- Line 161: The authors are asked to clarify "In some experiment"
Sentence was modified as “In some experiments (Figure 5D-E),”
- Line 429: showed similar RSV replication kinetic than adults and were able to produce - perhaps change to "to"; ".. similar .. to ..".
Sentence was corrected as suggested.

Reviewer 2 Report
In this paper, Laubreton et al. investigated the impact of nBreg on lung innate immunity and the crosstalk with AMs upon RSV infection. Their experiments are well designed and the presented data is very clear and supports their hypothesis. Still, I have some small remarks and questions:
- For the normalization of qPCR data it is recommended to use between two and five validated stably expressed reference genes for normalization, as advised in Vandesompele et al. 2002. Did the authors first perform a pilot study to determine the most stable housekeeping genes? Why was only GAPDH used for normalization?
- Regarding statistical significance, it would be better to define *,** and *** and ****. In the legend of Figure 1, the authors describe (*** P<0.001) but not the * and ** shown in the figure. Always add the used asterisks in the figure legends.
- Can the authors comment on the fact that there is no data in Figure 2A for the 14 day old mice? Based on the pictures, I would think that the infection at day 4 is less pronounced in 14 day old mice than in adult mice. I was curious what the results of day 5 where in 14 day old mice and how they relate to the data of day 4 in 14 day old or day 5 in adults.
- Please clarify the following lines: Interestingly, twelve days-old neonates had low CD5+ nBreg percentages close to those of adult animals (Figure 1A). Thus, between 6 and 14 days of life, mice became capable to contain the viral replication at early d.p.i. and developed an anti-viral response confirming an age-dependent production of IFN-I in RSV infection (line 283-285).
- Lines 301-302: Neonatal AMs produced significant quantities of IFN-a and IFN-b in response to RSV infection (Figure 4B). What do the authors mean by signficant quantities? Compare to what? Please rephrase
- Line 306: The authors mention that they have never detected IL-10 production (date not shown). Can they elaborate on the different experimental set-ups they have tried?
- Figure 4A: For the expression levels of IFN-a and b, Have the authors considered looking at earlier timepoints? IFN responses are very early, especially on RNA levels
- Lines 358-360: IL-10 production by CD5+ nBreg was increased in the presence of conditioned medium (Figure 5D) while RSV replication, measured by the level of the viral N-RSV RNA expression was reduced (Figure 5E). In my opinion, this conclusion cannot be made based on the data shown in Figure 5. There are only two data points and the variation is very high. Please soften this conclusion or repeat the experiment to obtain more data points.
- Is it possible to use IFNAR deficient mice to instead of IFNAR antibody?
- The presented flow cytometry data nicely shows that there are no nBreg cells in the lungs of adult mice. Can the authors hypothesize, or do they know what happens with the nBreg cells. Do they die, evolve to another phenotype or relocate?
- Supplemental Figure 1A: Is it possible to name the selected populations on the plots? For example, nBreg, immature B cells,.. This will make it easier to follow as a reader.
Author Response
- For the normalization of qPCR data it is recommended to use between two and five validated stably expressed reference genes for normalization, as advised in Vandesompele et al. 2002. Did the authors first perform a pilot study to determine the most stable housekeeping genes? Why was only GAPDH used for normalization?
Analysis were also performed using HPRT as a housekeeping gene, and similar results were obtained. Thus it is possible to normalize qPCR data of alveolar macrophages using GAPDH and HPRT. However, in order to obtain a minimum of RNA material to practice the reverse transcriptase, our cellular tests were performed with 50,000 cells per well, which requires at least the euthanasia of 3-4 neonatal mice per well. We have therefore chosen to standardize our pPCR data on a single housekeeping gene to allow maximum testing on other genes and thus, indirectly, to minimize the use of animals.
- Regarding statistical significance, it would be better to define *,** and *** and ****. In the legend of Figure 1, the authors describe (*** P<0.001) but not the * and ** shown in the figure. Always add the used asterisks in the figure legends.
Figures are corrected as suggested.
- Can the authors comment on the fact that there is no data in Figure 2A for the 14 day old mice? Based on the pictures, I would think that the infection at day 4 is less pronounced in 14 day old mice than in adult mice. I was curious what the results of day 5 where in 14 day old mice and how they relate to the data of day 4 in 14 day old or day 5 in adults.
The experiment with 14-days-old mice was conceptualized to illustrate the loss of early permissiveness of neonates to RSV infection. In this perspective of studying early infection and in an effort to limit the use of animals in the experiments, we did not plan analyses at 5 days post-infection, a time when lung replication disappears in both adults and 6-days-old mice.
- Please clarify the following lines: Interestingly, twelve days-old neonates had low CD5+ nBreg percentages close to those of adult animals (Figure 1A). Thus, between 6 and 14 days of life, mice became capable to contain the viral replication at early d.p.i. and developed an anti-viral response confirming an age-dependent production of IFN-I in RSV infection (line 283-285).
Sentence was modified as “Thus, between 6 and 14 days of life, mice became capable to contain the viral replication at early d.p.i. and at the same period of life started to produce small amount of IFN-b, confirming an age-dependent anti-viral response to RSV infection [8, 9]. Interestingly, this age period was associated with a decrease in CD5+ nBreg numbers in the lung of neonate, to reach numbers close to those of adults (Figure 1A).”
- Lines 301-302: Neonatal AMs produced significant quantities of IFN-a and IFN-b in response to RSV infection (Figure 4B). What do the authors mean by significant quantities? Compare to what? Please rephrase
Sentence was modified as “Neonatal AMs expressed IFN-a1 and IFN-b mRNA (Figure 4A) and produced detectable quantities of IFN-a and IFN-b in response to RSV infection (Figure 4B), however these IFN-I responses were significantly lower than in adult AMs (Figure 4B).” Moreover, an indicator “#” has been added to the figure (new figure 4) to show significant differences between control conditions and viral infection condition.
- Line 306: The authors mention that they have never detected IL-10 production (date not shown). Can they elaborate on the different experimental set-ups they have tried?
Sentence was modified as “our experimental set-up for detection of inflammatory cytokines (Figure 4), we never detected IL-10 production by RSV-infected neonatal and adult AMs (data not shown, luminex on supernatant and qPCR on cell lysates).”
- Figure 4A: For the expression levels of IFN-a and b, Have the authors considered looking at earlier timepoints? IFN responses are very early, especially on RNA levels
It is an interesting suggestion, however we did not do it. Our cellular tests required at least 50,000 cells per well, which implicate at least the euthanasia of 3-4 neonatal mice per well. We have therefore tested only one time-point post-viral exposure to analyse the IFN-I pathway.
- Lines 358-360: IL-10 production by CD5+ nBreg was increased in the presence of conditioned medium (Figure 5D) while RSV replication, measured by the level of the viral N-RSV RNA expression was reduced (Figure 5E). In my opinion, this conclusion cannot be made based on the data shown in Figure 5. There are only two data points and the variation is very high. Please soften this conclusion or repeat the experiment to obtain more data points.
Sentence was modified as “IL-10 production by CD5+ nBreg tended to increase in the presence of conditioned medium (Figure 5D) while RSV replication, measured by the level of the viral N-RSV RNA expression tended to diminish, although statistical analysis could not be performed due to small number of experimental data points”.
- Is it possible to use IFNAR deficient mice to instead of IFNAR antibody?
To our knowledge, the IFNAR1 knock-out mice exist only on the C57Bl/6J or 129S2/SvPas genetic backgrounds, while we used BALB/c mice to perform our studies. It is known that RSV infection differs between BALB/c mice and C57Bl/6J mice, the later strain being less sensitive to RSV infection (Tregoning JS, Yamaguchi Y, Wang B, et al. Genetic susceptibility to the delayed sequelae of neonatal respiratory syncytial virus infection is MHC dependent. J Immunol. 2010;185(9):5384-5391. doi:10.4049/jimmunol.1001594.). Thus we made the choice to keep the BALB/c genetic background and use anti-IFNAR antibodies to block the IFN-I response.
- The presented flow cytometry data nicely shows that there are no nBreg cells in the lungs of adult mice. Can the authors hypothesize, or do they know what happens with the nBreg cells. Do they die, evolve to another phenotype or relocate?
This question currently remains open as there is no publication specifying the fate of these cells. We make the following hypothesis, which remains to be confirmed.
IL-33 was showed to promote a subset of IL-10-producing Breg-like cells in vivo (Sattler et al., 2014). It has recently been shown that IL-33 was produced following the first breathes, the lung epithelium also produces high amount of IL-33, and this contributes to the recruitment of innate cells such as ILC2 (De Kleer et al., 2016; Saluzzo et al., 2017). It could thus be hypothesized that this IL-33 production by lung epithelium could contribute to the CD5+ nBreg recruitment in the lungs during the neonatal period.
It was shown with human Breg that CD5 promotes Breg survival by stimulating autocrine IL-10 production. Since we observed a correlation between the level expression of CD5 at the surface of CD5+ nBregs (measured by the mean of fluorescence intensity by flow cytometry) and the number of CD5+ nBreg in the lungs, it could be hypothesized that the level of CD5 expression modulates the number of CD5+ nBreg.
References:
Sattler S, Ling GS, Xu D, Hussaarts L, Romaine A, Zhao H, Fossati-Jimack L, Malik T, Cook HT, Botto M, Lau YL, Smits HH, Liew FY, Huang FP. IL-10-producing regulatory B cells induced by IL-33 (Breg(IL-33)) effectively attenuate mucosal inflammatory responses in the gut. J Autoimmun. 2014 May;50(100):107-22. doi: 10.1016/j.jaut.2014.01.032. Epub 2014 Feb 1. PMID: 24491821; PMCID: PMC4012142.
- de Kleer, M. Kool, M. J. de Bruijn et al., “Perinatal activation of the interleukin-33 pathway promotes type 2 immunity in the developing lung,” Immunity, vol. 45, no. 6, pp. 1285–1298, 2016.
- Saluzzo, A. D. Gorki, B. M. J. Rana et al., “First- breath-induced type 2 pathways shape the lung immune environment,” Cell Reports, vol. 18, no. 8, pp. 1893–1905, 2017.
Gary-Gouy, H., et al., Human CD5 promotes B-cell survival through stimulation of autocrine IL-10 production. Blood, 2002. 100(13): P. 4537-43.
- Supplemental Figure 1A: Is it possible to name the selected populations on the plots? For example, nBreg, immature B cells,.. This will make it easier to follow as a reader.
Colors and names were added on the plots and legend was modify as follow “Conventional (CD23+ CD5-, red square), immature (CD23- CD5-, green square) and CD5+ B-cells (blue square) were separated according to their CD23 and CD5 expression.”
